# Compact and Lightweight Mid-IR Laser Spectrometer for Balloon-borne Water Vapor Measurements in the UTLS

Manuel Graf[1,3], Philipp Scheidegger[1,2], André Kupferschmid[2], Herbert Looser[1], Thomas Peter[3], Ruud Dirksen[4], Lukas Emmenegger[1], and Béla Tuzson[1]

[1]Laboratory for Air Pollution/Environmental Technology, Empa - Swiss Federal Laboratory for Materials Science and Technology, 8600 Dübendorf, Switzerland

[2]Transport at Nanoscale Interfaces, Empa - Swiss Federal Laboratory for Materials Science and Technology, 8600 Dübendorf, Switzerland

[3]Institute for Atmospheric and Climate Science, ETH Zürich, 8092 Zürich, Switzerland

[4]Deutscher Wetterdienst (DWD)/GCOS Reference Upper Air Network (GRUAN) Lead Center, Lindenberg, Germany

**Correspondence:** Béla Tuzson (bela.tuzson@empa.ch)

**Abstract.** We describe the development, characterization, and first field deployments of a quantum cascade laser direct absorption spectrometer (QCLAS) for water vapor measurements in the upper troposphere and lower stratosphere (UTLS). The instrument is sufficiently small ($30 \times 23 \times 11\,\mathrm{cm}^3$) and lightweight ($3.9\,\mathrm{kg}$) to be carried by meteorological balloons and used for frequent soundings in the UTLS. The spectrometer is a fully independent system, operating autonomously for the duration of a balloon flight. To achieve the required robustness, while satisfying stringent mass limitations, the concepts for optics and electronics have been fundamentally reconsidered compared to laboratory-based spectrometers. A significant enhancement of the mechanical and optical stability is achieved by integrating a newly designed segmented circular multipass cell which allows for $6\,\mathrm{m}$ optical path length in a very compact fashion. The $H_2O$ volume mixing ratio is retrieved by calibration-free evaluation of the spectral data, i.e., only relying on SI-traceable measurements and absorption line parameters. The open-path design reduces the risk of contamination, allows fast response, and thus high vertical resolution. Laboratory-based characterization experiments show an agreement within $2\,\%$ to reference measurements and a precision of $0.1\,\%$ under conditions comparable to the UTLS. The instrument successfully performed two balloon-borne test flights up to $28\,\mathrm{km}$ altitude. In the troposphere, the retrieved spectroscopic data show an excellent agreement with the accompanying measurements by a frost point hygrometer (CFH). At higher altitude, the quality of the spectral data remained unchanged, but outgassed water vapor within the instrument enclosure was hindering an accurate measurement of the atmospheric water vapor. Despite this limitation, these test flights demonstrated the operation of a compact laser spectrometer in the UTLS aboard a low-volume meteorological balloon, opening the perspective for future highly resolved, accurate, and cost-efficient soundings.

## 1 Introduction

The atmospheric water vapor concentration is an important climate variable and plays a crucial role in various processes from the ground to the upper atmosphere. It affects the Earth's radiative balance, not only in the lower troposphere, but also in the upper troposphere and lower stratosphere (UTLS) (Harries, 1997; Held and Soden, 2000; Solomon et al., 2010; Dessler et al.,

2013). The strong ability of water vapor to absorb infrared radiation makes it the largest contributor to the natural greenhouse effect. At the same time, water vapor is a prerequisite for the formation of clouds, which scatter incoming solar shortwave radiation back to space, thereby increasing the Earth's albedo. In addition, $H_2O$ is the major source of the OH radical in the stratosphere (Hanisco et al., 2001), making it a relevant player in UTLS chemistry. Climate modeling and projections rely on the accurate knowledge of the water vapor abundance throughout the lower and middle atmosphere and on the understanding of the underlying processes that control its spatio-temporal variability. This requires frequent, accurate, and vertically highly resolved water vapor concentration measurements, which cannot be achieved by remote sensing techniques alone. While the in-situ humidity measurement near the Earth's surface is a well-established procedure (e.g. using capacitive sensors), obtaining accurate and reliable values in the cold and dry UTLS is incomparably more challenging.

The detection of the multiannual trends of UTLS water vapor concentration is only possible on the basis of highly precise and long-term comparable measurements. In addition, the investigation of currently open questions in the field of cloud microphysics (Peter et al., 2006) require a high accuracy and SI-traceability. Therefore, next generation UTLS hygrometers should ideally feature the following specifications:

($i$) High accuracy ($< 10\,\%$) in the UTLS, i.e. $8$–$20\,km$ altitude at an effective spatial resolution of a few meters, corresponding to measurements at $1\,Hz$ or faster;

($ii$) SI-traceability to ensure long-term comparability and to avoid calibration-induced errors;

($iii$) Sufficient compactness and low weight to enable flexible and frequent deployment aboard of ordinary meteorological balloons without the need for special permission, i.e. a total mass of $4.5\,kg$ or less, depending on the local legislation.

The development of an instrument that simultaneously fulfills all these requirements has proven very difficult, as can be deduced from the significant efforts over the past half century. The required accuracy and weight constraints seem mutually exclusive under the demanding environmental conditions, especially for spectroscopic techniques. In addition, the risk of water vapor self-contamination, owed to the strong adhesive properties of the $H_2O$ molecule, has been a limiting factor particularly for large instrumentation (Rollins et al., 2014).

Currently, cryogenic frost point hygrometers (CFH/FPH) (Vömel et al., 2016; Hall et al., 2016) are the most established instruments for high-accuracy humidity measurements in the UTLS, however, these are yet themselves not free of uncertainties (Brunamonti et al., 2018; Jorge et al., 2020). Additionally, these devices require a fundamental reconception due to their use of fluoroform (R23, $CHF_3$) as cooling agent, which must be phased out according to the Montreal Protocol, because of its high global warming potential (UNEP, 2016). Alternative frost point hygrometers based on thermoelectric cooling (TEC) are currently under development (Sugidachi, 2018; Jorge, 2019).

Beside the CFH approach, numerous spectroscopy-based techniques have been deployed in the UTLS in the past three decades, e.g. Lyman-$\alpha$ photofragment fluorescence, tunable diode laser, and also mass spectrometry (SPARC, 2000). However, intercomparison campaigns have revealed disagreements that exceed the stated accuracy of the individual instruments (Rollins et al., 2014; Fahey et al., 2014). In addition, all these devices suffer from deployment-related limitations, being either aircraft-based (Silver and Hovde, 1994; Sonnenfroh et al., 1998; May, 1998; Diskin et al., 2002; Zondlo et al., 2010; Dyroff et al.,

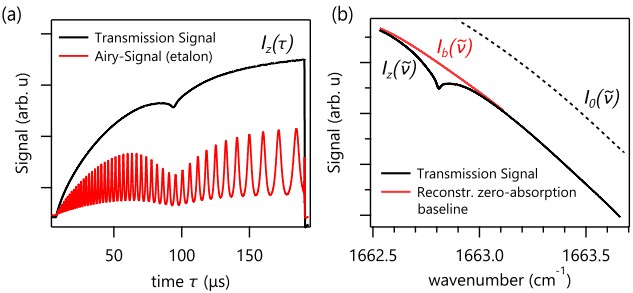

**Figure 1.** (a) The raw transmission signal through air (black) and a Fabry-Pérot-etalon (red). The Airy signal is used to determine the frequency tuning of the laser. (b) After time-to-frequency conversion, the absorption line is fitted onto a subset of the data. Thereby, the baseline $I_b(\tilde{\nu})$ is reconstructed, which, in contrast to the laser output intensity $I_0(\tilde{\nu})$, includes broadband absorption losses.

2015; Thornberry et al., 2015; Buchholz and Ebert, 2018), requiring large balloon gondolae (Durry et al., 2005; Moreau et al., 2005; Gurlit et al., 2005; Durry et al., 2008), or being restricted to nighttime measurements (Lykov et al., 2011).

In many respects, laser absorption spectroscopy is the method of choice for high-accuracy and contact-free trace gas mea-surements (Hodgkinson and Tatam, 2013). The advantages lie in the fast acquisition rate, the high molecular specificity and
the possibility of calibration-free measurements (Buchholz and Ebert, 2018; Hunsmann et al., 2008), i.e., measurements that are only based on physical parameters such as pressure, temperature and molecule specific spectroscopic data. However, the transition from the laboratory to a balloon-borne instrument is ambitious as it requires a reconsideration of the established concepts. Herein, we describe the development of the first miniaturized, fully standalone mid-IR laser absorption spectrometer for accurate in-situ measurements of water vapor in the UTLS carried by low-volume meteorological balloons.

**2   Methods**

**2.1   Optical principles**

The principle of tunable laser direct absorption spectroscopy relies on the excitation of ro-vibrational states of molecules by a narrow-band light source. In this application, we exploit the fast tunability of a quantum cascade laser to record transmission spectra at kHz repetition rate that cover several wavenumbers. From these spectra, the number density of target molecules is
deduced using the Beer-Lambert-law (Eq. 1), which describes the attenuation of radiation in absorbing media. The residual intensity $I_z(\tilde{\nu})$ of a laser beam at wavenumber $\tilde{\nu}$ after passing an optical path length (OPL) $z$ through a homogeneous absorbing medium is expressed by

$$I_z(\tilde{\nu}) = I_0(\tilde{\nu})A(\tilde{\nu},t)\exp(-nz\sigma(\tilde{\nu})), \tag{1}$$

where $n = N/\tilde{V}$ denotes the number density, i.e., the number of absorbing molecules $N$ within a volume $\tilde{V}$. The wavenumber
dependence of the absorption coefficient $\sigma(\tilde{\nu})$ can be approximated by a Voigt profile $V(\tilde{\nu}; \alpha_D(T), \gamma(p,T))$. The broadening

parameters $\alpha_D$ and $\gamma$ are calculated as a function of the gas pressure $p$ and temperature $T$ using the coefficients from a spectroscopic database. $A(\tilde{\nu}, t)$ represents the cumulative broadband transmission losses, e.g., due to windows, finite reflectivity of mirrors, or the scattering by dust particles or aerosols. This factor may vary over the course of multiple measurements; therefore, it is determined for each spectrum individually. In fact, $AI_0(\tilde{\nu}) = I_b(\tilde{\nu})$ corresponds to the expected signal in absence

of absorbing target molecules (Fig. 1a), hereafter referred to as 'baseline'. In laboratory-based closed-cell spectrometers, $I_b(\tilde{\nu})$ can be regularly determined by removing the target species from the measurement volume, e.g., by purging the absorption cell with 'zero-air' or evacuation. Since this is impossible for the herein used open-path system, $I_b(\tilde{\nu})$ is reconstructed during the evaluation procedure. Due to the non-linear laser output intensity during the frequency scan (Fig. 1(b)), $I_b(\tilde{\nu})$ is approximated by a polynomial $P_N(\tilde{\nu})$ of the order three to six, depending on the considered spectral range. Since the absorption line shape

is fully defined by the absorption coefficient $\sigma(\tilde{\nu})$, the number density $n$ of a particular absorbing species can be determined by minimizing $X$, i.e. the squared differences between the acquired signal $I_z(\tilde{\nu}_i)$ and the model function. Thereby,

$$X(n, P_N) = \sum_{i=0}^{m} [I_z(\tilde{\nu}_i) - P_N(\tilde{\nu}_i) \exp(-nz\sigma(\tilde{\nu}_i))]^2 \tag{2}$$

is minimized over the $m$ points of the spectrum under variation of $n$ and the polynomial coefficients $p_1 \ldots p_N$ of $P_N = \sum_{j=0}^{N} p_j \tilde{\nu}^j$. Clearly, the reconstruction of $P_N$ becomes more accurate if the dataset includes sections far from the line cen-

90 ter, i.e., where $\sigma(\tilde{\nu}) \approx 0$. The minimization of $X$ is performed using the Levenberg-Marquardt least-squares algorithm (Press et al., 2007). The spectral line intensity and the broadening parameters are taken from the HITRAN2016 database (Gordon et al., 2017), whereas the actual gas pressure $p$ and temperature $T$ are measured. It is important to note that Eq. 1 establishes a well-defined relation between the (unknown) number density $n$ and the (measured/reconstructed) absorbance $\ln(I_z/I_b)$. The relation only contains directly measurable quantities ($T, p, z, I_z$) and molecular properties ($\sigma$). This renders direct absorption

spectroscopy a potentially calibration-free method to determine the number density of a trace gas. The ideal gas law can be applied to convert the number density $n$ into an amount fraction $\chi = n/n_{\text{tot}}$ in units of $\text{mol} \, \text{mol}^{-1}$, where $n_{\text{tot}}$ denotes the number density of all present molecules. Calling $\chi$ a volume mixing ratio and using *parts-per* notation is a widely accepted practice in the field. For simplicity, we follow this terminology throughout this paper.

In this work, we used the Voigt function to approximate the spectral absorption coefficient $\sigma(\tilde{\nu})$ lack of available parameters

for more accurate, higher order profiles. However, the collected data can be reprocessed with more sophisticated line shape models once the corresponding parameters are determined.

Figure 1(a) illustrates a typical raw transmission signal $I_z(\tau)$ as a function of time $\tau$ using a quantum cascade laser (QCL) as a light source. As a consequence of resistive heating during a pulse of driving current, the QCL gradually changes its emission wavelength. This spectral sweeping, however, neither happens at constant tuning speed nor at constant emission intensity.

Therefore, the tuning characteristics $d\tilde{\nu}/dt$ must be determined specifically for each driving configuration and operating temperature. This is achieved by recording the Airy transmission signal of a Fabry-Pérot etalon, which yields intensity oscillations with equidistant maxima in wavenumber space. Thereby, the time axis of the raw signal is converted into a wavenumber axis (Fig. 1(b)).

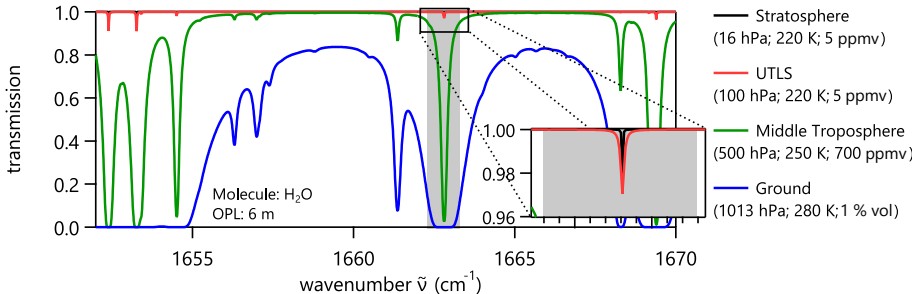

**Figure 2.** Simulated transmission spectra of $H_2O$ in the mid-IR (around $6\,\mu m$) at several vertical locations of the atmosphere for an absorption path length of $6\,m$. The selected absorption line is distinctly isolated such that the zero-absorption baseline remains accessible for a broad range of pressures and $H_2O$ concentrations. Its intensity is strong enough to ensure good SNR even at stratospheric conditions. The grayed part denotes the tuning range of the QCL.

## 2.2 Absorption line selection

A crucial point in the design of a laser absorption spectrometer is the accurate detection of the absorption signal $I_z/I_b$, which usually consists of a tiny absorption feature on top of a large background signal. Therefore, the signal-to-noise-ratio (SNR) must be sufficiently high to allow the precise determination of the absorption, even at the lowest encountered abundance. The strength of the absorption signal scales with the optical path length (OPL). The OPL is usually extended using multipass optics, which is often the size-determining component within a spectrometer. In addition, the absorption signal can be optimized by the

selection of a proper wavelength range that contains a suitable absorption line. Many small inorganic molecules exhibit their strongest fundamental transitions in the mid-IR spectral region. $H_2O$ exhibits an absorption band associated to the bending vibration mode ($\nu_2$) around $1600\,cm^{-1}$ ($6\,\mu m$) which is about $30\,\%$ stronger than the absorption band due to the symmetric ($\nu_1$) and asymmetric stretching modes ($\nu_3$) around $3700\,cm^{-1}$ ($2.7\,\mu m$) and about 10 times stronger than the first $\nu_2 + \nu_3$ combination band at around $5300\,cm^{-1}$ ($1.9\,\mu m$). To exploit this advantage and thereby reduce the required OPL, we are using

a QCL, i.e. fast tunable and powerful semiconductor mid-IR light sources (Faist et al., 1994). The target wavelength region has been determined by scanning the $\nu_2$-vibration band of water vapor, searching for interference-free absorption lines with highest dynamic range under the conditions found during a balloon flight to the UTLS. The selected spectral window is shown in Fig. 2 highlighted in gray, which also corresponds to the spectral coverage of the QCL tuned by current. This window is advantageous since it contains a strong and isolated absorption line ($2_{21} \leftarrow 2_{12}$ at $1662.809\,cm^{-1}$) which facilitates the access to baseline $I_b$

in its vicinity. According to the HITRAN-database, the absorption parameters for this line, especially the intensity, are known with an accuracy better than $2\,\%$ (Birk and Wagner, 2012; Gamache and Hartmann, 2004; Lodi et al., 2011; Toth).

## 2.3 Instrumentation

### 2.3.1 Optical layout

The selected absorption line (Fig. 2) is preferably combined with an OPL of $6\,\mathrm{m}$ to ensure an SNR above 20 even under
UTLS conditions. Such an optical path length enhancement can be achieved with a variety of multipass cells (MPCs). How-
ever, the choice of the MPC is critical for optical and mechanical performance: In fact, there is typically a trade-off between
well-controlled and interference-free beam folding, compactness, and mechanical stability. We addressed this trade-off by the
development of the segmented circular multipass cell (SC-MPC) (Graf et al., 2018), which is schematically shown in Fig. 3(a).
This monolithic cell consists of a rotationally symmetric arrangement of individual mirror segments carved into its inner sur-
face. This makes the SC-MPC highly resistant to thermally induced distortion, while the spherical shape of the segments
preserves a confined laser beam even upon multiple reflection. In addition, the SC-MPC geometry is well-suited for open-path
applications since the air can stream through the optical plane perpendicularly without any obstruction.

The SC-MPC designed for this instrument weighs $160\,\mathrm{g}$ and contains 57 segments ($6 \times 6\,\mathrm{mm}^2$) which are circularly arranged
with a diagonal distance of $108.82\,\mathrm{mm}$. SC-MPCs are generally tolerant to various input beam shapes. In addition, the last
segment is specifically curved such that the laser beam is directly focused onto the detector. Therefore, the IR detector and
the laser housing can be attached in immediate vicinity to the MPC without the need of additional beam-shaping optics. This
enables an extremely compact setup, increases the mechanical stability, and reduces the optical path outside the sampling
area. These key features of the SC-MPC renders it suitable also for more generic applications in mobile laser spectrometry, as
illustrated by Tuzson et al. (2020) with drone based methane detection.

As a light source, we use a distributed feedback quantum cascade laser (DFB-QCL) packaged in a TEC-equipped HHL
housing with embedded collimation optics (Alpes Lasers SA, Switzerland). It is operated at a base temperature of $24\,°\mathrm{C}$.

### 2.3.2 Integration

The instrument is built around this highly compact optical layout, which is shown in Fig. 3(a). It is realized in an open-path
configuration, such that the air can freely stream through the central funnel with a minimal diameter of 86 mm (Fig. 3(b)). This
yields a large flow rate of $30\,\mathrm{l\,s^{-1}}$ upon $5\,\mathrm{m\,s^{-1}}$ ascent rate, which helps reducing the influence of self-contamination caused by
desorbing water vapor. The central funnel is extended externally by a duct of $10\,\mathrm{cm}$ length (PTFE), preventing contaminated
air from the proximity of the instrument enclosure to stream through the measurement zone. PTFE is chosen because of its
low porosity and its low outgassing rate under reduced pressure (Weissler and Carlson, 1980). In addition, it is hydrophobic
and non-adhesive, thus preventing the deposition of hydrometeors or condensate on its surface. Flexible bellows, which tightly
connect the MPC with the instrument enclosure, inhibit the propagation of external mechanical stress onto the optical system,
while suppressing the convective exchange of cold outside air with the internal volume.

For thermal reasons, the highly temperature sensitive laser and detector are not in direct contact with the MPC, which is
fully exposed to the external air temperature variation. The laser is mounted on a custom-made aluminum alignment stage
that allows high-precision positioning along five axes. Having all necessary degrees of freedom covered at the laser side, the

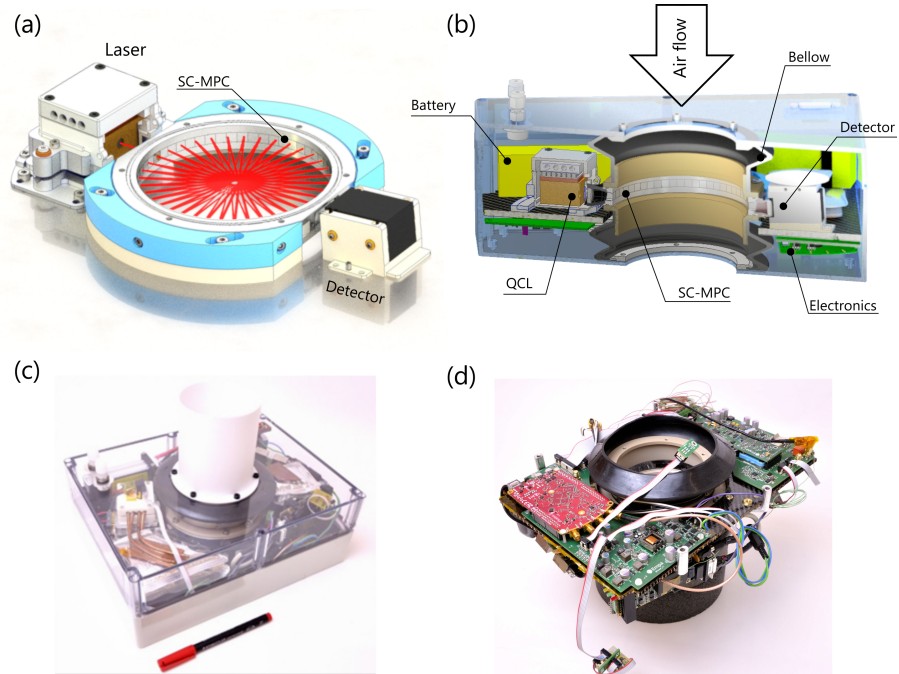

**Figure 3.** (a) Highly compact optical layout using the segmented circular MPC (SC-MPC), which enables the mounting of laser and detector very close to its coupling apertures. The beam reflection pattern is schematically depicted by the red trace. (b) Virtual cut showing the integration of our optical concept within the fully assembled spectrometer in open-path configuration. (c) Photo of the fully assembled spectrometer without insulation. (d) Photo of the driving and data acquisition electronics mounted below the optical system.

mounting of the detector can be kept simple, since the beam alignment is accomplished by adjustments of the laser. Thus, the detector is enclosed by a 3D printed holder that is directly attached to the lightweight carbon-aramid honeycomb base plate. The detector holder, which further acts as thermal insulation, is covered by an aluminum plate that is connected to heat pipes and serves as a heat exchanger. This efficient and compact construction permits the optical plane to be located only $1.75\,\mathrm{cm}$ above the surface of the base plate. At this height, the MPC is held in position by 3D printed braces, which are equipped with a heating wire allowing the temperature-control of the cell, e.g., to prevent icing or condensation. The entire instrument is enclosed in a polycarbonate box, as shown in Fig. 3(c). The lid contains two connections for purging the internal volume with dry gas prior to lift-off. In combination with a $4\,\mathrm{cm}$ thick insulating layer of expanded polystyrol (XPS) added around the enclosure, the entire instrument weighs $3.86\,\mathrm{kg}$.

### 2.3.3 Thermal management

Thermal control and stabilization are of utmost importance for high-precision laser spectrometers. The large temperature variation encountered during a balloon flight renders this especially challenging. To avoid laser frequency drifts, the temperature of the QCL must remain within a few $\mathrm{mK}$, while the atmospheric air temperature may change by $80\,\mathrm{K}$. In addition, the electron-

ics' excess heat ($\sim 15\,\mathrm{W}$) must be managed. This cannot be done by passive coupling to the surrounding atmosphere, because of the large temperature span, which would lead to uncontrollable changes in cooling power. In fact, passive coupling would require a variation of the heat transfer coefficient by about one order of magnitude during ascent, in order to keep the heat source at a constant temperature: On the ground and at altitudes above ca. $20\,\mathrm{km}$, efficient cooling is required, while insulation is needed during the rest of the flight to maintain the internal temperature.

Our strategy of thermal management relies on the fact that the instrument has to be stabilized only for a limited amount of time ($\approx 2\,\mathrm{h}$), and the direction of heat flow is reversed during flight. These are ideal conditions for the use of a heat buffer. More specifically, the instrument is thermally decoupled from the external temperature by the insulating XPS layer mounted around the enclosure. Sensitive electronic elements are additionally insulated except at one well-defined area of heat exchange. This area is connected, either directly or via heat pipes, to an organic phase-change material (PCM, RT 18 HC, Rubitherm, Germany) as buffer medium with a phase transition at $T_c = 18\,^{\circ}\mathrm{C}$, which is close to the operating temperature of the laser. This concept adds passive, nonlinear, and bidirectional thermal inertia. With a combined heat capacity of $260\,\mathrm{kJ\,kg^{-1}}$ (i.e., $72.2\,\mathrm{W\,h\,kg^{-1}}$; latent plus specific heat between $11\,^{\circ}\mathrm{C}$ and $26\,^{\circ}\mathrm{C}$) an amount of $416\,\mathrm{g}$ PCM is required to fully take up $15\,\mathrm{W}$ during $2\,\mathrm{h}$. Assuming a bidirectional use, the amount of PCM can be halved. Depending on heat production and tolerable temperature range, the individual electronic components are equipped with $10$–$40\,\mathrm{g}$ of PCM, encapsulated in custom-made pads. The laser, as the most sensitive device, is additionally stabilized by a PID-controlled TEC, while its heat sink is connected to the buffer medium. This combination of active control and increased thermal inertia successfully limits the variation of the internal air temperature during a balloon flight to $T_0 \pm 10\,\mathrm{K}$ and the laser chip to $\Delta T = 19\,\mathrm{mK}$ as shown in Fig. 4. The laser temperature variation is derived by its frequency shift, which is determined using the absorption line position as a reference (red trace). At the tropopause, where the line contrast is lowest, i.e., the absorption line is still broad but has a rather small amplitude; the determination of the central frequency is most difficult and thus exhibits the largest noise level.

### 2.3.4 Laser driving and data acquisition

Most of the custom-developed electronic circuitry boards (PCBs) are integrated on the bottom side of the carbon base plate, as indicated in Fig. 3(d). The core of the electronics system is given by a commercial single board computer 'Red Pitaya' (STEMLab, Slovenia; partly open source). It features a system on chip (SoC) that combines a field programmable gate array (FPGA) and a microcontroller unit (MCU). The FPGA has been reconfigured to provide the functionalities for high-resolution absorption spectroscopy (Liu et al., 2018; Tuzson et al., 2020).

The rapid spectral sweeping of the the QCL is achieved by periodic modulation of the laser driving current. An especially economic strategy is referred to as 'intermittent continuous wave' (ICW) (Fischer et al., 2014), whereby the driving current is applied in pulses, followed by a moment of complete shut-down of the laser to re-establish its initial temperature. In comparison to the generic continuous wave (CW) driving schemes, ICW driving drastically reduces the energy consumption, and thus the production of excess heat. The current pulses are generated by custom-developed analogue electronics (Liu et al., 2018). For the detection of the transmission intensity signal we use a thermoelectrically cooled MCT-detector (PVM-2TE-8 $1\times1$, Vigo System, Poland) coupled to a small-footprint preamplifier (SIP-DC-20M) with a bandwidth of $20\,\mathrm{MHz}$.

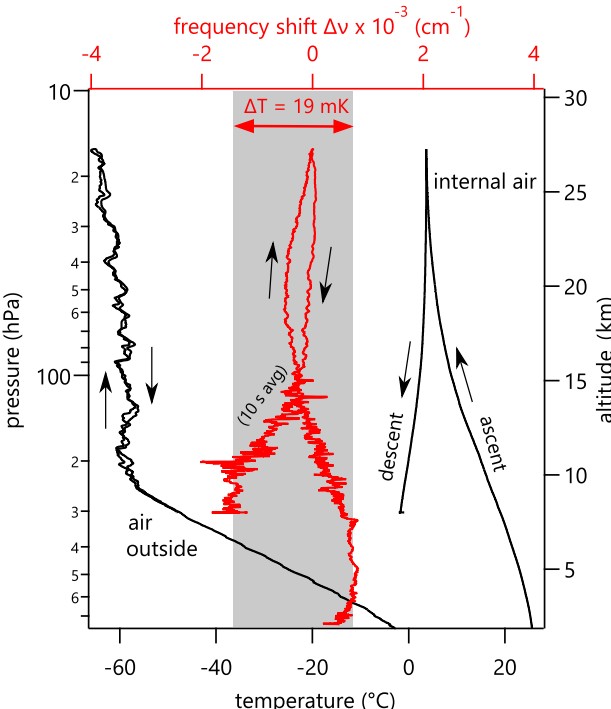

**Figure 4.** Temperature measurements during the first balloon flight in the UTLS demonstrating the thermal stability of the instrument. The laser temperature variation is derived from its detuning, i.e., the change of the absorption line's central frequency (red, top axis), indicating a maximal temperature drift of $19\,\mathrm{mK}$. The internal and external air temperatures are shown in black on the bottom axis for comparison. The pronounced variability of the laser temperature around the tropopause (300–100 hPa) is caused by the fitting procedure of the central frequency of the absorption line rather than effective temperature changes.

The dataflow within the instrument is summarized in Fig. 5. The preamplified raw signal is digitized by a 14-bit analog-digital-converter (ADC) at $125\,\mathrm{MS\,s^{-1}}$ that is integrated on Red Pitaya. The FPGA provides the trigger signals for the laser driver such that the data acquisition is synchronized: Each current pulse creates one individual spectrum. In our case, the current pulse lasts for $200\,\mu\mathrm{s}$, consists of 25'000 data points, and is followed by a shut-off period of $100\,\mu\mathrm{s}$, resulting in laser duty cycle of $67\,\%$. To improve the SNR of the signal, the FPGA sums up 3000 individual spectra in realtime, leading to an effective measurement rate of 1 Hz. The number of averaged spectra and the duty cycle can, however, be specifically selected, taking into account the trade-off between precision, temporal resolution, and covered wavelength range.

After completion of the spectral summation, the dataset is transferred to the MCU using the internal RAM in a ring-buffer scheme. These data can either be sent to a remote computer via a TCP/IP-interface or stored on a flash memory for post-processing. In addition to the fast acquisition of spectra, various sensors and status variables are accessed at 1 Hz for spectral or diagnostic purposes. The communication, the processing, and the storage of these values is handled by Python-based scripts

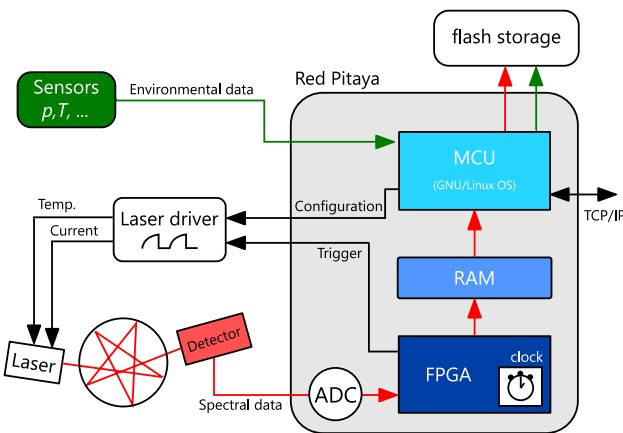

**Figure 5.** Schematics of data acquisition and transfer within the instrument. To acquire spectral data, the laser driver is triggered by the FPGA and generates current pulses of predefined length, amplitude, and shape. A multitude of additional sensors both for spectroscopic as well as diagnostic purposes (green) are regularly read out. The internal clock of the FPGA synchronizes the different tasks.

running on the single board computer. This concept allows easy implementation of additional features such as controlled heating of the MPC, managing the laser temperature, reading out an accelerometer, etc.

## 3 Validation experiments

### 3.1 Laboratory-based

The characterization and validation of the spectrometer at representative conditions in the laboratory is highly demanding, given the wide span of temperature, pressure, and water vapor concentration encountered during balloon flights. More specifically, volume mixing ratios $\chi$ of $10\,\mathrm{ppmv}$ and less are very challenging to produce accurately due to the pronounced adsorption properties of $H_2O$. Similarly, temperatures of $-70\,°C$ and pressures as low as $100\,\mathrm{hPa}$ can only be obtained in dedicated climate chambers. Since it was not possible to cover the entire space of occurring $p$, $T$, and $\chi$ in a well-controlled fashion, a subspace of these target conditions was assessed in individual experiments. To characterize the performance at low pressure, the instrument is exposed in open-path configuration to specific combinations of $p$ and $\chi$ at room temperature in a home-built pressure chamber. The chamber is continuously purged by $H_2O/N_2$ mixtures, gravimetrically admixed using a dynamic generator (HCD311, HovaCAL, IAS GmbH, Germany), as depicted in Fig. 6(c). Due to the large surface area of the chamber walls, the lowest achievable $H_2O$ mixing ratio is $2\,‰vol$. To obtain an absorbance representative to the UTLS under these conditions, the measurements are taken at a neighboring absorption line at $\nu_0 = 1665.7\,\mathrm{cm}^{-1}$, which exhibits about three orders of magnitude weaker absorption intensity (cf. Fig. 6(a)).

In a first experiment, the precision of the instrument at constant conditions is assessed using the Allan-Werle-deviation technique (Werle et al., 1993), as shown in Fig. 6(b), where the evolution of precision as a function of the averaging time is

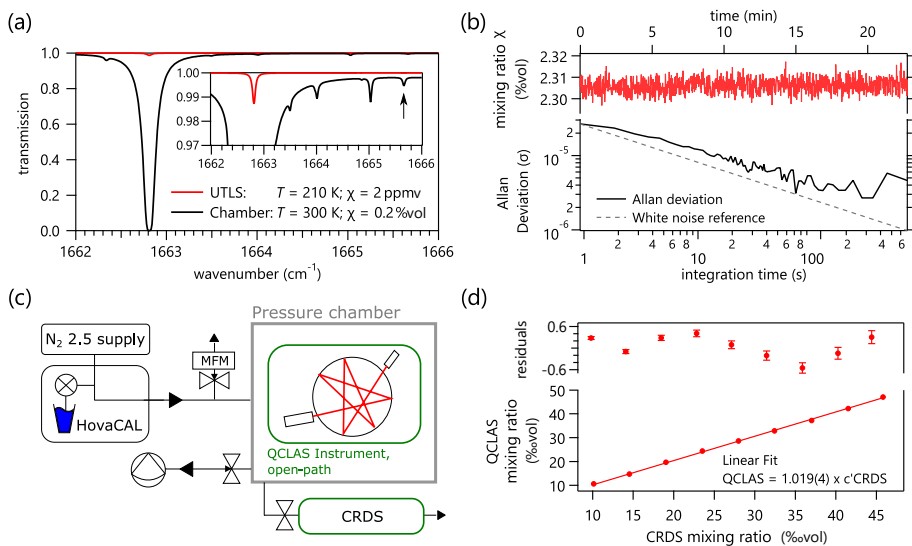

**Figure 6.** (a) Absorption lines in the vicinity of the one selected for UTLS measurements are about three orders of magnitude weaker, which renders them a representative alternative for measurements at high $H_2O$ concentration. (b) Allan-Werle-deviation plot at $259\,\mathrm{hPa}$ for an absorbance representative for the UTLS. A relative Allan deviation of $0.11\,\%$ at $1\,\mathrm{Hz}$ is found, showing further enhancement upon longer averaging times. (c) The laboratory-based setup for open-path instrument characterization. (d) Direct comparison of the calibration-free QCLAS results with a calibrated CRDS system at $253\,\mathrm{hPa}$, showing excellent agreement with a coefficient of determination $R^2 = 0.9989$.

quantified. For this measurement, the sampling volume is held at $\chi = 2.3\,\%\mathrm{vol}$, $p = 258.5\,\mathrm{hPa}$, and $T = 296\,\mathrm{K}$ by continuous purging. To ensure stable operation of the instrument for longer period than the targeted $2\,\mathrm{h}$, an externally driven liquid cooling system is used (UC180, Solid State Cooling Systems, USA). Values for the mixing ratio $\chi$ are acquired at $1\,\mathrm{Hz}$, after co-averaging of 3000 individual spectra. An Allan-Werle-deviation of $26\,\mathrm{ppmv}$ ($0.11\,\%$) is obtained at an integration time of $1\,\mathrm{s}$,
while another order of magnitude improvement on precision can be achieved by integration over hundred seconds.

In a second experimental series, the accuracy of the instrument is assessed by comparison to the measurements of a commercially available cavity ring-down spectrometer (CRDS; Model G2401, Picarro Inc, USA). The CRDS analyzes extracted air from the chamber within its built-in pressure-controlled cavity. This sampling cell is held at $200\,\mathrm{hPa}$, thereby defining the lower end of accessible pressures. Figure 6(d) shows excellent agreement between the QCLAS and the calibrated CRDS at $253\,\mathrm{hPa}$.
Calibration of the CRDS was based on a dew-point hygrometer, while the QCLAS results are found by a calibration-free procedure, as discussed above.

Additional tests assessing the mechanical and thermal sensitivity have been performed, elaborating the performance of the instrument under field conditions. This includes climate chamber experiments at stratospheric pressure, temperature, and humidity; however, due to technical limitations, without the possibility of controlling more than one of these parameters
simultaneously. A detailed discussion of these experiments is given in Graf (2020).

## 3.2 In-flight test and intercomparison

As a first assessment of the novel QCLAS instrument under realistic conditions, two test flights were performed at Meteorological Observatory Lindenberg, Germany, which also hosts the lead center of the GCOS Reference Upper-Air Network (GRUAN) (Bodeker et al., 2016). The QCLAS was deployed in the UTLS on the 17th and 18th of December 2019. The instrument was attached with a $45\,\text{m}$ rope to a small volume balloon (TX1200, $3\,\text{m}^3$) and reached altitudes of $28.3\,\text{km}$ and $27.4\,\text{km}$, respectively, on the two consecutive days. Both deployments were accompanied by a parallel ascent of a CFH, serving as reference for the measured $H_2O$ volume mixing ratios. The CFH was launched on a separate balloon to keep the payload below $4\,\text{kg}$. However, the starts were sufficiently close in time to make CFH-QCLAS comparisons meaningful; CFH launches just $37\,\text{min}$ and $7\,\text{min}$ before the QCLAS for Flight I and Flight II, respectively. During Flight II, the CFH data showed a clear and strong contamination-associated offset, especially above the tropopause, after ascending through a $800\,\text{m}$ thick layer with $RH > 98.5\,\%$. Therefore, we focus here on the evaluation and comparison of the first flight, where the complete ascent and descent datasets are available from both instruments. For the spectroscopic retrieval of the $H_2O$ mixing ratios, the $p$ and $T$ values are used from an attached RS41 radiosonde (Vaisala, Finnland). Taking these values from the standardized and well-characterized radiosonde is preferred over the use of the system-integrated sensors, since temperature measurements are highly delicate under these conditions and strongly depend on the specific integration properties of the $T$-sensor (Shimizu and Hasebe, 2010). The descent data stops as soon as the ground station has lost the signal to the RS41, even though the QCLAS continues to operate down to $5\,\text{km}$ altitude. The instrument was recovered using the GPS coordinates received from the locator module connected to the radiosonde. The spectra were recorded at a repetition period of $300\,\mu\text{s}$, which is a good trade-off between broad spectral coverage and scanning speed as it enables the co-averaging of 3000 spectra to reach $1\,\text{Hz}$ temporal resolution. This corresponds to a vertical spatial resolution of $5\,\text{m}$ considering the ascending speed of the balloon.

### 3.2.1 Results and discussion

Figure 7(a) shows the vertical profile of $H_2O$ mixing ratios from both instruments for ascending and descending measurements over the available range of altitudes. Figure 7(b) focuses on the troposphere, where the data from the QCLAS are in excellent agreement with the CFH. The mean deviation is only $3\,\%$ and the values remain close even where the absorption line is already partly saturated. In the right panel of this figure, the relative deviation is plotted, showing a standard deviation of $13\,\%$ between 2 and $10\,\text{km}$ altitude. Large differences are observed around sudden changes in the humidity profile caused by natural short-scale spatio-temporal variations in the tropospheric moisture. The deviations between both instruments can be explained by the slightly different flight trajectories (Fig. 7(c)) as well as the faster response time of the QCLAS compared to the CFH.

Above ca. $10\,\text{km}$ altitude, the QCLAS gradually retrieves higher $H_2O$ mixing ratios compared to the CFH. This is attributed to the enhanced outgassing of water vapor from the internal surfaces of the instrument over the course of the flight. The outgassed water vapor interacts with the laser beam inside the enclosure of the instrument within the short optical path ($9\,\text{mm}$) between the laser/detector and the cell entrance. Although this internal OPL is only $0.14\,\%$ of the total OPL, the corresponding absorption can still generate a significant offset in the data with increasing altitude. For example, a superposition of the atmo-

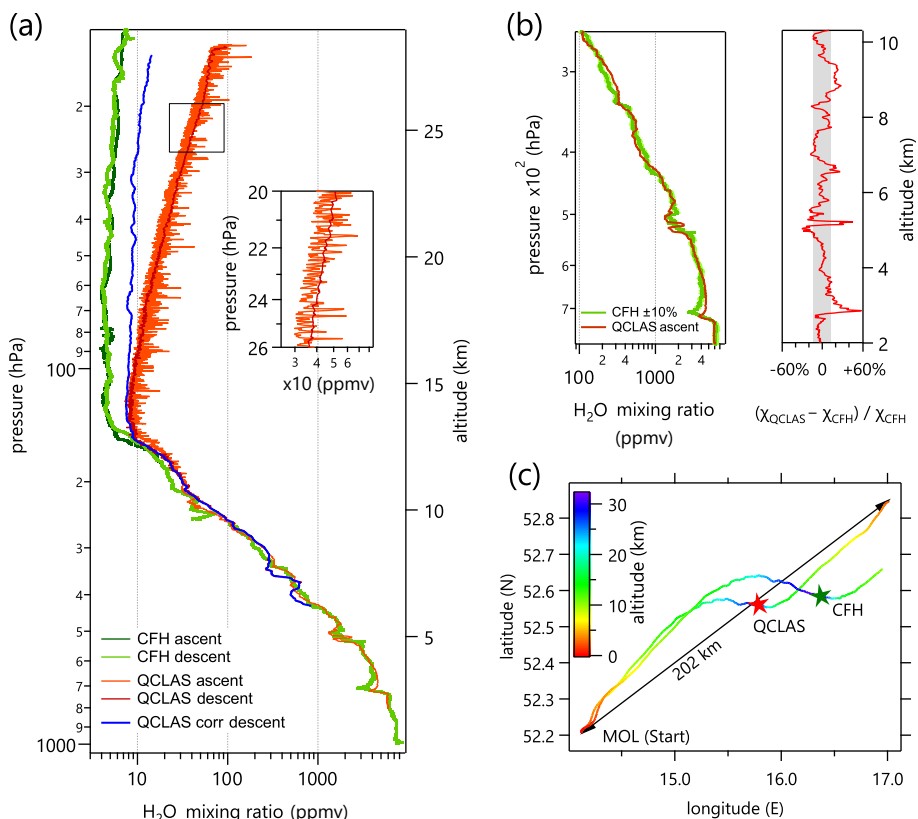

**Figure 7.** (a) $H_2O$ volume mixing ratio profiles measured by QCLAS and CFH during ascent and descent. The inset shows the unilateral spikes on top of a slowly varying signal in the stratosphere. An attempt to correct for the spurious absorption caused the internal OPL is shown as blue curve (see main text for details). (b) Close-up of the tropospheric measurements; the relative deviation is plotted in the right panel, showing a standard deviation of 13 % (gray area) among the instruments. (c) Flight trajectories and burst points – the two balloons were launched with a time lag of 37 min.

spheric absorption due to 5 ppmv water vapor at 20 km altitude (50 hPa) and the internal absorption by $10^4$ ppmv $H_2O$ over
0.14 % of the total OPL at the same pressure, results in an apparent mixing ratio of 19 ppmv. This bias could not be avoided
even by thoroughly flushing the instrument with dry nitrogen prior to lift-off. The dominant effect of internally trapped water
vapor is also the reason for the slightly higher apparent concentration at descent than at ascent in the stratosphere: The internal
concentration continues to increase during the first 3 km of descent and remains higher (cf. Fig. 7(a) and inset). An attempt to
correct this bias by measuring the internal humidity with a low-cost capacitive sensor is shown as the blue curve for the QCLAS
descent data. For better comparison to CFH, a 20 s moving average filter is applied to this curve. While this correction partly
removes the trend, the sensor fails to deliver plausible absolute values in the stratosphere, where the environmental conditions
(p < 200 hPa, RH < 8 %) are far beyond the specifications for this capacitive RH-sensor. Therefore, it is likely that the remaining
bias between CFH and the corrected QCLAS data is caused by a significant error of the internal humidity measurement.

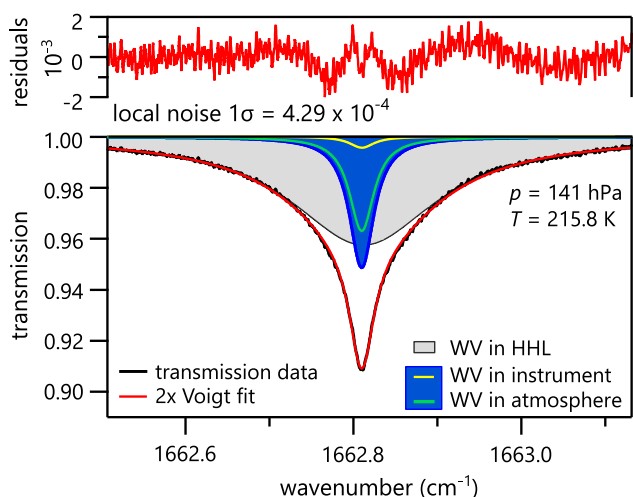

**Figure 8.** Representative transmission spectrum recorded during balloon-borne deployment at $13.8\,\text{km}$ altitude. The spectrum exhibits two spectroscopically distinguishable features: a broad absorption (gray) due to the enclosed water vapor within the laser housing HHL (OPL of $3\,\text{mm}$) at ca. $1000\,\text{hPa}$ and a narrow absorption (blue) at a pressure of $141\,\text{hPa}$. The latter feature is again a superposition of two contributions: the atmospheric absorption (green) within the MPC (OPL of $6\,\text{m}$ and $T = 215.8\,\text{K}$) as well as the absorption by trapped water vapor within the instrument's enclosure, which is not distinguishable spectroscopically (yellow).

This problem of internal outgassing is therefore to be addressed in the near future by a technical adaptation: Either by completely eliminating the internal optical path or by establishing a pressure difference within the instrument with respect to the surrounding atmosphere. The latter may be achieved by guiding the laser beam through leak-tight channels that cover the internal OPL sections and allow to maintain substantially higher pressure within these compartments during flight. This would enable the spectroscopic disentanglement of the absorption features due to the different pressure broadening.

The strategy of distinguishing different line contributions according to their pressure broadening is currently applied successfully to discriminate the contribution of residual humidity within the housing of the laser: Figure 8 shows an in-flight spectrum recorded at the hygropause ($13.8\,\text{km}$). The narrow absorption feature due the atmospheric and instrument-internal humidity at low pressure ($141\,\text{hPa}$) is superimposed over a broad feature (gray) that originates from residual moisture within the HHL housing of the laser device. The distance between the laser chip facet and the window of the laser housing amounts to approximately $3\,\text{mm}$. Because of the leak-tight sealing of the laser enclosure, the pressure remains constant during flight, therefore allowing the precise disentanglement of these contributions due to their unequal pressure broadening, especially at high altitude. In the evaluation procedure, we account for this effect by the simultaneous fitting of two absorption line profiles at their corresponding pressure.

Apart from the offset due to internal humidity, the mixing ratio data during ascent shows distinct spikes with an amplitude of up to $40\,\text{ppmv}$ on top of a slowly varying signal above the tropopause at about $10\,\text{km}$ altitude. A close-up of this structure is presented as an inset in Fig. 7(a). A Fourier transform analysis indicates a preferred occurrence of these spikes with periodicities

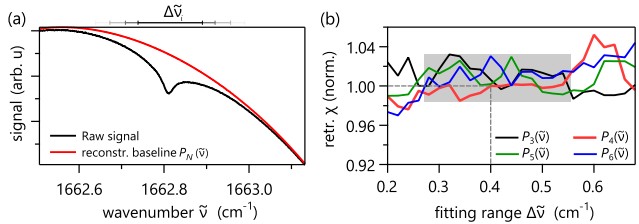

**Figure 9.** (a) Fitting of the measured transmission data. The choice of the baseline polynomial (red) and the selected range $\Delta\tilde{\nu}_i$ to evaluate a measured spectrum need to be optimized. Varying these parameters can influence the retrieved $H_2O$ mixing ratio $\chi$, as shown in (b) after repeated evaluation of the same spectrum. Using polynomial orders $P_N(\tilde{\nu})$ between 3 and 6, the standard deviation of all fitting results within the gray highlighted region amounts to 1.1 % relative to the finally selected configuration for the analysis of the data, i.e. $\Delta\tilde{\nu} = 0.4\,\mathrm{cm}^{-1}$ and $P_4(\tilde{\nu})$.

of $t_1 = 6.5\,\mathrm{s}$ and $t_2 = 12.4\,\mathrm{s} \approx 2t_1$. Assuming a gravitational pendulum as an idealized model describing the instrument's motion below the balloon, the expected periodicity would be $t_p = 2\pi\sqrt{L/g} = 13.5\,\mathrm{s}$ for the used rope length $L = 45\,\mathrm{m}$. The accordance of the calculated periodicity with the GPS data makes it plausible that these spikes are caused by repeated transitions of the instrument through the contaminated wake of the balloon in an oscillating fashion. This hypothesis is strongly supported by the fact that the spikes immediately disappear after the burst of the balloon, indicating that they are balloon-associated rather than originating from the enclosure of the instrument. Possible reasons for the absence of a similar effect in the CFH data are the lower temporal resolution and the longer rope ($60\,\mathrm{m}$) for the CFH. The quantitative contribution of these individual factors is difficult to estimate, but will be investigated during future test flights of the QCLAS. Most importantly, however, any balloon-related contamination can be avoided by measuring during descent (Kräuchi et al., 2016), which is perfectly feasible due to our instrument's high gas exchange rate and its high temporal resolution capabilities.

### 3.2.2 Uncertainty assessment

In the UTLS, the biases discussed above inhibit the assessment of the accuracy by direct comparison to CFH. Therefore, to quantify the expected accuracy of our spectrometer, we compiled a bottom-up uncertainty budget and discuss the relevant contributions. These sources of uncertainty may be grouped into four categories:

($i$) **Spurious water vapor (external and internal)**: These contributions are of technical nature as discussed in the previous section. They are to be avoided by hardware adaptations on the instrument and the deployment system.

($ii$) **Uncertainty of environmental parameters**: The relative uncertainty of the $T$ and $p$ measurements by the RS41 radiosonde is specified to 0.2 % and 0.5 % (Vaisala), respectively.

($iii$) **Thermally induced variations**: The thermal expansion/contraction of the MPC during the flight, due to a temperature variation of 54 K, induces a change of 0.12 % in the OPL. Since thermal expansion is a simple function of temperature, this effect could, in principle, be included in the evaluation procedure. However, this small effect was not taken into consideration during the data processing in this case. In contrast to the thermal expansion, variations of the laser temperature can lead to

changes that are more difficult to quantify. Although the thermal management system successfully stabilizes the laser temperature within 19 mK during the entire flight (cf. Fig. 4), the residual change in the laser temperature still causes a frequency shift of $3 \times 10^{-3}\,\mathrm{cm}^{-1}$. This may lead to a variation of the laser's tuning characteristics that were determined prior the flight using a Fabry-Pérot etalon. Corresponding calculations indicate a maximum change of 0.76 % in the derived mixing ratio during the entire flight.

($iv$) **Uncertainties of the spectroscopic retrieval**: For the retrieval of the water vapor mixing ratio based on the recorded spectra, an important source of uncertainty is the limited accuracy of the line parameters given in the spectral database, i.e., 1–2 % on the line intensity and 1 % on the pressure-broadening coefficient. For comparison, the normalized spectral noise level ($1\sigma$) of $4.3 \times 10^{-4}$ has a negligible contribution to the total uncertainty, i.e., 0.1 % at 1 s as indicated by the Allan-deviation plot. However, the retrieved concentrations show a slight dependence on the fitting configuration, i.e., the spectral window size $\Delta\tilde{\nu}$ and the chosen polynomial order $N$, used for baseline reconstruction (cf. Fig. 9(a)). In order to quantify the sensitivity of the retrieved concentration on the selection of $\Delta\tilde{\nu}$ and $P_N(\tilde{\nu})$, the same spectrum is evaluated with different fitting configurations, as shown in Fig. 9(b). This comparison is performed for spectra at different altitudes. As expected, the variation of the mixing ratio $\chi$ among different fitting presets is highest when the contrast of the absorption line is low, i.e. the height of the absorption feature divided by its width is small. This is particularly the case at the hygropause, where the ambient pressure is still relatively high, while the $H_2O$ abundance is low (cf. Fig. 8). Nevertheless, even for low contrast spectra, the deviations remain small within a reasonable range of fitting configurations. The evaluated mixing ratios $\chi$ reach a plateau in the region $0.28 < \Delta\tilde{\nu} < 0.55\,\mathrm{cm}^{-1}$ for baseline polynomials of the order 3 to 6. Within this region (red area), the calculated $\chi$ show a standard deviation of only 1.1 %. Therefore, the central values $\Delta\tilde{\nu} = 0.4\,\mathrm{cm}^{-1}$ and $P_4(\tilde{\nu})$ are the selected evaluation settings to analyze the presented experimental data. In Fig. 8(b), the mixing ratio retrieved with this configuration is set to unity. Other combinations within the red area show maximal deviations of +3.2 % and -1.2 %.

Based on the contributions discussed here, the total measurement-related uncertainty ($ii$-$iv$) amounts to 2.5 %. Additional comparison flights in the near future and the participation in intercomparsion campaigns will enable to experimentally verify this number and to further characterize the performance of the instrument. It should be noted that this budget does not include the potential error induced by using Voigt profiles, which do not describe ro-vibrational transitions in their full complexity (Ngo et al., 2013; Tran et al., 2013). This manifests itself as a characteristic W-shaped structure in the residuals (cf. Fig. 8) that even varies with pressure. For certain near-IR transitions, deviations of 2–5 % in the absorption line integral have been found between a simple Voigt and higher-order profiles (Lisak et al., 2015). However, these cases are chosen to illustrate the benefit of more elaborate line shape models and thus likely represent an upper limit of disagreement. These more sophisticated line shapes require up to four additional fitting parameters, which are currently not available for the molecular transition used here. Nevertheless, future laboratory-based experiments should include a detailed investigation of the herein selected absorption line to determine these missing parameters.

## 4 Conclusions

This work describes the development of a compact QCL-based direct absorption spectrometer with a total weight of merely 3.9 kg, which is designed for the measurement of water vapor in the UTLS aboard of meteorological balloons. It relies on an open-path segmented circular MPC and includes specifically developed hardware, such as low power dissipation laser driver electronics and FPGA-based data acquisition, as well as dedicated controlling software. A tailored thermal stabilization system based on a combination of phase-change material and thermoelectric cooling allows autonomous operation of the instrument during balloon-carried ascents and subsequent descents on a parachute. The open-path design minimizes self-contamination and enables a fast response time (<1 Hz), which is confirmed by the identification of individual wake transitions during balloon-borne ascent. Laboratory experiments show a precision of 0.1 % and excellent agreement with a CRDS instrument, supporting the calibration-free evaluation approach. A balloon-borne comparison to a CFH reveals a relative average deviation of only 3 % in the troposphere. The accuracy of the stratospheric measurements is currently limited by outgassing water vapor within the enclosure of the instrument, leading to a bias of the measured concentration at high altitude. Therefore, it is of highest importance to eliminate the influence of the internal optical path by a constructional adaptation. Once this source of error is removed, it will be possible to experimentally quantify the overall accuracy of the instrument by a direct comparison to CFH. Apart from the issue of internal outgassing, this lightweight and standalone mid-IR spectrometer showed highly stable operation even in the stratosphere, thus representing a promising candidate for future high-accuracy assessments of UTLS water vapor on a regular basis.

*Data availability.* The data used in this manuscript are available from the corresponding author upon request.

*Author contributions.* M.G. designed and developed the instrument, collected and evaluated the data under the supervision of B.T. who managed the project together with L.E. and T.P. P.S. designed and developed the electronics hardware. A.K. developed and implemented FPGA and DAQ functionalities. H.L. developed analysis software. R.D. provided support to launch from Lindenberg. M.G. prepared the manuscript with contributions from all authors.

*Competing interests.* The authors declare that they have no conflict of interest.

*Acknowledgements.* The authors kindly acknowledge Badrudin Stanicki and Killian Brennan for their technical support, Curdin Flepp and Florian Lienhard for their contribution in software development. Further, many thanks to Tobias Bühlmann from METAS for his help with the laboratory characterization experiments. Special thanks to the GRUAN team at DWD Lindenberg, in particular Sabine Körner, for their support with the balloon launches and instrument recovery. We also thank Erwin Pieper and Urs Hintermüller from the mechanical

engineering workshop at Empa for their support and the helpful discussions. This project has received funding from the Swiss National Science Foundation (SNSF) under grant agreement nr. 157208.

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
