# Peer review of "Compact and Lightweight Mid-IR Laser Spectrometer for Balloon-borne Water Vapor Measurements in the UTLS"

_Atmospheric Measurement Techniques, 2020_

## Referee Comment (RC1) · Anonymous Referee #1 · 16 Aug 2020

The paper "Compact and Lightweight Mid-IR Laser Spectrometer for Balloon-borne Water Vapor Measurements in the UTLS" of Graf et al. describes the development, characterization and test of a new lightweight balloon-borne instrument for water vapour measurements in the UTLS region. The compact instrument employs mid-IR absorption spectroscopy as operating principle, with a quantum cascade laser as a light source and a segmented circular multipass absorption cell for path length augmentation.

New developments of lightweight and balloon borne instruments for the challenging measurements of water vapour in the UTLS region are interesting, in particular if they

have the potential to establish SI traceability and do not rely on cryogens that are threatened by phase out such as $CHF_3$, on which current frost point hygrometers depend.

The paper is thus highly significant and it fits very well with the scientific scope of AMT. The discussion is clear, scientifically sound, and well argumented. The overall presentation is also clear, with only a few Figures requiring slight adjustments to enhance the readability.

There are only two minor remarks that apply to the paper in its current form, which should be published after minor corrections. First, the capability of measuring at high temporal resolution is pointed out at several instances all over the paper and in the abstract. However, apart from the better than 1 Hz requirement and the capability to measure at 1 s time resolution, no clear number is given with respect to the time resolution reached by the final instrument. I would suggest putting such a number (or a qualifier such as sub-Hz) in the abstract and the conclusion to make things clear.

Second, an uncertainty budget (L301+) for the water vapour measurements is provided, but the discussion does not include any of the systematic biases that were discusssed in the previous paragraphs. While degassing from the balloon seems to be an unwanted effect that can be overcome using a longer rope or by measuring in descent, the problem of degassing into the 9 mm interior optical path has not yet been overcome by the present design. According to Fig. 7, this problem is apparent in the $> 13$ km altitude range, where a correction is applied that brings the water mixing ratio to constant values. What is the uncertainty of the measurements in this altitude range ? Then, there is (another ?) apparent offset ($> 10$ %) between QCLAS and CFH in the 11 to 13.5 km range. Where does this come from ? I could not find an adequate discussion, but obviously the proposed degassing correction does not account for the discrepancy. This systematic bias also warrants discussion in the text and a corresponding number should appear in the uncertainty budget.

[Figure]

Technical

L45 "these are yet itself ..." → "these are yet themselves ..."

L46 Mention the Kigali (2016) amendment to the Montreal Protocol.

L66 - 67 "The tunability of the laser allows to record entire transmission spectra at fast scanning rates, from which the number of molecules can be deduced." An entire rovi-brational spectrum of $\sim 100\,\mathrm{cm^{-1}}$ is hardly accessible for a conventional high-resolution laser. The sentence should be rephrased accordingly.

L109 "while the absorption signal can be optimized by the selection a wavelength range with strong" → "while the absorption signal can be optimized by the selection of a wavelength range with strong" or "while the absorption signal can be optimized by selecting a wavelength range with strong"

L112 "due to the (a)symmetric stretching mode ($\nu_1$ and $\nu_3$) ... ". Use of singular is confusing here. Better write, "due to the symmetric ($\nu_1$) and asymmetric ($\nu_3$) stretching modes ..."

L119 "According to the HITRAN-database, the absorption parameters for this line, especially the intensity, are known with an accuracy better than 2 %." While the reference to HITRAN is correct, the authors should also cite the original work that has led to the entry in HITRAN. Otherwise, the people who did the (spectroscopy) work don't get the credit.

L211 The SI recommended µmol / mol is to be preferred over ppmv, especially because SI recommended abundance (mixing) ratios are used elsewhere in the article (e.g. Figure 6).

L218 "mixing ratios is" → "mixing ratio is"

L233 - 235 "Numerous additional tests ... A detailed discussion of these experiments

can be fond in Graf (2020)" It is not quite clear how relevant the information is. Either the phrases should be omitted or the authors should give a very short summary/conclusion of the relevance of these tests for the current paper.

L256 - 261 "In the right panel of this figure, the relative deviation is plotted, showing a standard deviation of 13 % between 2 and 10 km altitude." The way the paragraph is organised is confusing, because the authors mention the standard deviation first and the mean deviation of 3 % only at the end. The order in which variables are presented should be reversed.

L327 "A standard deviation of 1.1 % relative to the finally chosen configuration ...". This sentence needs to be rephrased. It is not clear what "relative to" means in this context.

L335 "which are currently not available for the herein used molecular transition" → "which are currently not available for the molecular transition used here"

L336 "herein selected absorption line" → "selected absorption line"

L382 H2O - → "$H_2O$"

L442 H2O - → "$H_2O$"

L444 Camy-peyret → Camy-Peyret

Figure 6 labels are too small and hardly legible

Figure 8 Label identifies water vapour in "atmospheric." Should be "in atmosphere."

Figure 9 Graphs are too small. It is difficult to identify the different retrieved mixing ratios.

---

## Referee Comment (RC2) · Anonymous Referee #2 · 30 Nov 2020

The paper by Graf et al. "Compact and Lightweight Mid-IR Laser Spectrometer for Balloon-borne Water Vapor Measurements in the UTLS" describes the development and deployment of a balloon-borne laser spectrometer for water vapor measurements in the upper troposphere and lower stratosphere. The paper is generally well written. I have a few comments that should be addressed regarding some of the statements in the paper.

General comments:

It should be clearly stated that the development and testing of this sensor is far from complete if the authors claim measurement capability in the lower stratosphere. The

sensor must undergo much more detailed testing under LS conditions. The large humidity bias shown at below 10..15 ppm is inadequate for measurements in the lower stratosphere. The claim in the title that this sensor is for UTLS measurements is misleading. Many other research groups have spent incredible amounts of effort into sensor development, incl. multi-year inter comparison campaigns at climate chambers (cited in this manuscript).

Specific comments:

Title: I believe that the title (and paper) should not state that the presented instrument is for measurements in the UTLS, but rather for tropospheric soundings. The results presented do not warrant lower-stratospheric measurements.

P1, L9 The sentence "An open path design reduces the risk of contamination, allows fast response..." is contradictory of the results presented. Measurements presented here are indeed "contaminated".

P3, L67 ..the number of molecules within the light path can be..

P3, L73 which values from a spectroscopic database?

P4, L90 The integrated linestrength?

Fig. 2 indicate parameters such as pressure, concentration, altitude

Fig. 4 Explain in the main text what the problem of the fitting procedure was.

P10, L222 Fig. 6(a) should be Fig. 6(c)

P10, L231 the agreement of the slope is fine, but the intercept is really important, too. Especially when the goal of this instrument is to accurately measure extremely low $H_2O$. This is probably difficult to transfer from the weak absorption line probed for this test to the strong one due to different baseline shape, which might highlight a weak point of this experiment.

Fig. 6 "A precision of 0.11 % at 1Hz is.." 0.11 % of what? Also Rˆ2 = 1.018 should either state the slope or the correlation coefficient I assume.

P13, L278 Figure 8(a) should be Figure 8

P16, L351 No, you were not successful to measure LS water vapor.
* * *

---

## Author Comment (AC1) · 27 Dec 2020

**Authors' Response to Referee Comments #1**

We would like to thank the Referee for the constructive comments and helpful suggestions on the manuscript, which helped us to further improve the clarity of the paper. Below, we give detailed responses (in blue) where appropriate.

**General comments:** There are only two minor remarks that apply to the paper in its current form, which should be published after minor corrections. First, the capability of measuring at high temporal resolution is pointed out at several instances all over the paper and in the abstract. However, apart from the better than 1 Hz requirement and the capability to measure at 1 s time resolution, no clear number is given with respect to the time resolution reached by the final instrument. I would suggest putting such a number (or a qualifier such as sub-Hz) in the abstract and the conclusion to make things clear.

In the context of balloon-borne soundings, a few meters vertical distance per measurement are considered as high resolution. There are two requirements for such a rate to be reached: Not only the capability of a fast sampling guarantees high effective resolution; also, the air exchange rate in the sampling zone must be adequate. Therefore, we realized an open path sampling system that allows a high throughput of air. In addition, the recording of individual spectra (full scans) takes place at a rate of 3 kHz. These single spectra are then co-averaged 3000 times to improve the signal-to-noise ratio (SNR) and give roughly one humidity value per second. In principle, though, the number of co-averages can be reduced, leading to higher temporal resolution, but on the expense of precision. Thus, technically, the effective acquisition rate may be increased up to 3000 individual humidity measurements per second.

We agree that this should be mentioned more clearly in the manuscript. Therefore, we added additional information to the Section 2.3.4 Laser driving and data acquisition (P8, L205): "An individual spectrum is recorded during the applied current ramp that lasts for 200 us and consists of 25'000 measured intensity points. A dedicated FPGA functionality sums up in real-time a predefined amount of individual spectra -- usually a few thousand spectra per second -- in order to improve the SNR. The number of averaged spectra and the duty cycle can be individually selected, taking into account the trade-off between precision, temporal resolution, and covered wavelength range.

In addition, we add the following sentence to the methods description subsection of the balloon-borne experiment:

"The spectra were recorded at a repetition period of 300 us, which is a good trade-off between broad spectral coverage and speed, as it enables the co-averaging of 3000 spectra to reach 1 Hz temporal and 5 m vertical resolution."

Second, an uncertainty budget (L301+) for the water vapour measurements is provided, but the discussion does not include any of the systematic biases that were discussed in the previous paragraphs. While degassing from the balloon seems to be an unwanted effect that can be overcome using a longer rope or by measuring in descent, the problem of degassing into the 9 mm interior optical path has not yet been overcome by the present design. According to Fig. 7, this problem is apparent in the>13km altitude range, where a correction is applied that brings the water mixing ratio to constant values. What is the uncertainty of the measurements in this altitude range? Then, there is (another?) apparent offset (>10%) between QCLAS and CFH in the 11 to13.5 km range. Where does this come from? I could not find an adequate discussion, but obviously, the proposed degassing correction does not account for the discrepancy. This systematic bias also warrants discussion in the text and a corresponding number should appear in the uncertainty budget.

We agree that the discussion of uncertainty contributions is somewhat confusing. Therefore, we modified the Section 3.2.1 to make it clearer, discussing the reasons for the recorded undesired water absorption (1) within the instrument enclosure, (2) within the laser housing, and (3) due to balloon-related desorption in three different paragraphs. In this section, we also mention the attempt of correcting for contribution (1) and (2). Concerning (1), the outgassing of humidity within the instrument, the actual design does not yet offer a solution to the problem, indeed. There are ongoing efforts for providing a technical solution, though. During the reported measurements, the only option to correct for this offset was to subtract the estimated internal humidity contribution, based on the measurement of the embedded capacitance RHsensor. However, the conditions in question (<200 hPa, <8% RH) were far outside the specifications of this device, even yielding negative values for a relevant fraction of the measurement period. We have tried several strategies to calibrate this sensor against the QCLAS data, but it is very likely that this sensor exhibits non-linear behavior, especially at very low RH. This would largely explain the offset that already appears at about 11 km, meaning that the capacitance sensor underestimates low RH. At the moment, this is the only way to correct for the signal due to internal humidity, although with little information about the quality of this correction. We state this fact more clearly in the text by reformulating the corresponding sentences: "While this correction partly removes the trend, the sensor fails to deliver plausible absolute values in the stratosphere, where the environmental conditions (p < 200 hPa, RH < 8%) are far beyond the specifications for this capacitive RH-sensor. Therefore, it is likely that the remaining bias between CFH and the corrected QCLAS data is caused by a significant error of the internal humidity measurement."

Having discussed these unwanted, avoidable, or external contributions to the uncertainty we have identified in the experiment, Section 3.2.2 focuses on intrinsic method- or instrument-related sources of uncertainty that remain even under optimized conditions. For the demanded completeness, we add an additional category (0) spurious water vapor (external and internal), although this is not the focus of this section.

**Technical:**

L45 "these are yet itself ..."  $\rightarrow$  "these are yet themselves ..."

**Done**

L46 Mention the Kigali (2016) amendment to the Montreal Protocol.

**Done**

L66 - 67 "The tunability of the laser allows to record entire transmission spectra at fast scanning rates, from which the number of molecules can be deduced." An entire rovibrational spectrum of ~100 cm-1 is hardly accessible for a conventional high-resolution laser. The sentence should be rephrased accordingly.

The term 'spectrum' was used here to distinguish from single wavelength measurements. For clarity, the sentence is rephrased: "In this application, we exploit the fast tunability of a quantum cascade laser to record transmission spectra at kHz repetition rate that cover a few wavenumbers. From these spectra, the number density of target molecules is deduced using the Beer-Lambert-law, which describes the attenuation of radiation in absorbing media."

L109 "while the absorption signal can be optimized by the selection a wavelength range with strong"  $\rightarrow$  "while the absorption signal can be optimized by the selection of a wavelength range with strong" or "while the absorption signal can be optimized by selecting a wavelength range with strong"

**Done**

L112 "due to the (a) symmetric stretching mode (v1 and v3) ... ". Use of singular is confusing here. Better write, "due to the symmetric (v1) and asymmetric (v3) stretching modes ..."

**Done**

L119 "According to the HITRAN-database, the absorption parameters for this line, especially the intensity, are known with an accuracy better than 2 %." While the reference to HITRAN is correct, the authors should also cite the original work that has led to the entry in HITRAN. Otherwise, the people who did the (spectroscopy) work don't get the credit.

Done, we now cite the original sources of the used parameters, i.e. self- and air-broadening, air-broadening temperature dependence, the spectral line intensity, and the line position. L211 The SI recommended µmol/mol is to be preferred over ppmv, especially because SI recommended abundance (mixing) ratios are used elsewhere in the article (e.g.Figure 6).

In our view, the correct terminology is "amount of substance fraction", given in mol/mol units. However, the scientific community in this field most frequently uses the term "(volume) mixing ratio" in units of ppmv (e.g. Vömel et al. (2016), Lossow et al. (2019), etc.). To avoid confusion we included a clear definition on L95 and we now consistently use the unit (u)mol/mol throughout the manuscript.

L218 "mixing ratios is"→"mixing ratio is"

Done

L233 - 235 "Numerous additional tests ... A detailed discussion of these experiments can be found in Graf (2020)" It is not quite clear how relevant the information is. Either the phrases should be omitted or the authors should give a very short summary/conclusion of the relevance of these tests for the current paper.

Rephrased: "Additional tests assessing the mechanical and thermal sensitivity have been performed, elaborating the performance of the instrument under field conditions. This includes climate chamber experiments at stratospheric pressure, temperature, and humidity; however, due to technical limitations, without the possibility of controlling more than one of these parameters simultaneously. A detailed discussion..."

L256 - 261 "In the right panel of this figure, the relative deviation is plotted, showing a standard deviation of 13 % between 2 and 10 km altitude." The way the paragraph is organised is confusing, because the authors mention the standard deviation first and the mean deviation of 3 % only at the end. The order in which variables are presented should be reversed.

The two sentences are reorganized. The 3 % mean deviation is now stated first in connection to the left panel of 7(b), mentioning the standard deviation of 13% later.

L327 "A standard deviation of 1.1 % relative to the finally chosen configuration ...". This sentence needs to be rephrased. It is not clear what "relative to" means in this context.

Revised sentence to: "The evaluated mixing ratios  $\chi$  reach a stable plateau in the region  $\Delta v = 0.4 \text{ cm}^{-1}$  for baseline polynomials of the order 3 to 6. Within this region (red area), the calculated  $\chi$  show a standard deviation of only 1.1%. Therefore, the central values  $\Delta v = 0.4 \text{ cm}^{-1}$  and  $P_4$  (v) are the selected evaluation settings to analyze the presented experimental data. In Fig. 9(b), the mixing ratio retrieved with this configuration is set to unity. Other combinations within the red area show maximal deviations of +3.2 % and -1.2 %.

L335 "which are currently not available for the herein used molecular transition"  $\rightarrow$  "which are currently not available for the molecular transition used here"

Done

L336 "herein selected absorption line"  $\rightarrow$  "selected absorption line" L382 H2O -  $\rightarrow$  "H2O"L442 H2O -  $\rightarrow$  "H2O"

Done

L444 Camy-peyret → Camy-Peyret

Done

Figure 6 labels are too small and hardly legible

Done

Figure 8 Label identifies water vapour in "atmospheric." Should be "in atmosphere."

Done

Figure 9 Graphs are too small. It is difficult to identify the different retrieved mixing ratios Done; plots enlarged.

---

## Author Comment (AC2) · 27 Dec 2020

**Authors' Response to Referee Comments #2**

We would like to thank the Referee for the comments and suggestions. Below, we give detailed responses (in blue) where appropriate.

**General comments:** It should be clearly stated that the development and testing of this sensor is far from complete if the authors claim measurement capability in the lower stratosphere. The sensor must undergo much more detailed testing under LS conditions. The large humidity bias shown at below 10..15 ppm is inadequate for measurements in the lower stratosphere. The claim in the title that this sensor is for UTLS measurements is misleading. Many other research groups have spent incredible amounts of effort into sensor development, incl. multi-year inter comparison campaigns at climate chambers (cited in this manuscript).

We are sorry that the Reviewer was misled by the title/abstract of our manuscript and we regret not meeting his/her expectations. However, the title highlights the key aspects of our manuscript: It presents a novel instrument that has been developed specifically for the application in the UTLS. In addition, it was deployed aboard a meteorological balloon to reach the lower stratosphere (28 km altitude). The instrument has been found fully operational during the entire flight (in both ascent and descent) and, despite being just a prototype instrument, it completed two consecutive flights within two days.

We fully acknowledge the significant efforts of other research groups and we are well aware of the highly challenging measurement situation, but exactly this supports the fact that our unique mid-IR instrument and its performance demonstrated under flight conditions represents a substantial scientific progress and thus fully justifies the title of our manuscript. We do not claim a final and completely characterized instrument, but rather communicate our development, findings, and observations, while clearly pointing out the current limitations. Again, we are grateful for the reviewer's concern and have rephrased the abstract to avoid any misleading of the reader: "*At higher altitude, the quality of the spectral data remained unchanged, but outgassed water vapor within the instrument enclosure was hindering an accurate measurement of the atmospheric water vapor. Despite this limitation, these test flights demonstrated the successful operation of a laser spectrometer in the UTLS aboard a low-volume meteorological balloon, with the perspective of future highly resolved, accurate and cost-efficient soundings.*"

We fully agree that this is not the end of the journey, but more effort has to be invested to validate the laser spectrometer e.g. in concerted intercomparison campaigns. These activities will hopefully lead to much better agreements with the reference method and minimize the observed biases. We are currently working on technically sound solutions, which will be implemented in the near future.

To avoid potential misunderstanding, we also removed the term "conclusive" in L237 P11, changing the sentence to: "*As a first assessment of the novel QCLAS instrument under realistic conditions, …*"

In addition, we add a separate subsection called "*Outlook*" at the end of the manuscript, containing the recommended further developments.

**Specific comments:** Title: I believe that the title (and paper) should not state that the presented instrument is for measurements in the UTLS, but rather for tropospheric soundings. The results presented do not warrant lower-stratospheric measurements.

The motivation for the instrumental development with its specific design was driven by the need, complexity, and importance of UTLS water vapor measurements, -- a need which is clearly shared by the reviewer. Our focus is on the description of the development, characterization and very first field-testing of the instrument. This justifies the title, especially since we transparently discuss the current limitations, their implication for future research, and potential solutions to the issues we faced.

The data presented in the paper were repeatedly recorded under UTLS conditions. Although we have identified a bias at high altitude, the system remained fully operational in the UTLS, and the quality of the retrieved absorption spectra remains comparable throughout the whole flight. This alone represents a formidable achievement that is worth to be communicated. Undoubtedly, further work is required to improve, characterize and fully establish the credibility of our instrument in the scientific community.

P1, L9 The sentence "An open path design reduces the risk of contamination, allows fast response..." is contradictory of the results presented. Measurements presented here are indeed "contaminated".

In our statement, we meant the intuitive fact that any enhancement of the volume-to-surface ratio in the detection volume, as well as any enhancement of the sample flow leads to a reduction of ad/desorption-caused biases of the measurement. In this context, an open-path arrangement is the best possible solution.

Indeed, our results do show a wet bias in the stratosphere. However, on P13, L261, the sources of this bias is discussed in detail, delivering profound arguments for being (1) balloon-caused and (2) related to the internal volume of the instrument (gap between laser/detector and cell), which is unrelated to the open-path construction of the sampling setup.

In the revised version, we try to emphasize the difference between these two aspects. They are again mentioned in the Subsection "*Outlook*" that is added to the revised manuscript.

P3, L67. the number of molecules within the light path can be..

These sentences are now rephrased for better understanding: "*In this application, we exploit the fast tunability of a quantum cascade laser to record transmission spectra at kHz repetition rate that cover several wavenumbers. From these spectra, the number density of target molecules is deduced using the Beer-Lambert-law, which describes the attenuation of radiation in absorbing media.*"

P3, L73 which values from a spectroscopic database?

For clarity, we rewrite the sentence at P3, L73: "*The wavenumber dependence of the absorption coefficient $\sigma(v)$ can be approximated by a Voigt profile $V(v;\alpha(T),\gamma(p,T))$. The broadening parameters $\alpha$ and $\gamma$ are calculated as a function of the gas pressure $p$ and temperature $T$ using the coefficients from a spectroscopic database.*"

The next sentence is moved towards the end of the paragraph for sake of structure.

P4, L90 The integrated line strength?

No change. According to the HITRAN naming convention, $S_{ij}$, i.e., the spectral integral of the absorption coefficient $\int \sigma \, dv = S_{ij}$ is called "spectral line intensity".

Fig. 2 indicate parameters such as pressure, concentration, altitude

Done.

Fig. 4 Explain in the main text what the problem of the fitting procedure was.

We added to the text at the end of Section 2.3.3: "*The laser temperature variation is derived by its frequency shift, using the absorption line position as reference (red trace). At the tropopause, where the line contrast is lowest, i.e., the absorption line is still broad but has a rather small amplitude; the determination of the central frequency is most difficult and thus exhibits the highest noise level.*"

P10, L222 Fig. 6(a) should be Fig. 6(c)

Corrected.

P10, L231 the agreement of the slope is fine, but the intercept is really important, too. Especially when the goal of this instrument is to accurately measure extremely low H2O. This is probably difficult to transfer from the weak absorption line probed for this test to the strong one due to different baseline shape, which might highlight a weak point of this experiment.

We agree that the intercept is important to qualify the accuracy. However, in our spectroscopic approach, assuming a zero intercept is well motivated. This is because the absence of target molecules can be determined with an uncertainty that is given by the Allan deviation. In our case, the uncertainty amounts to 0.01 % (for 100 s averaging), which can safely be considered as zero in this context.

Using a deliberately chosen small absorption line to determine the performance at relatively high concentrations may be surprising. However, it avoids the extreme challenges of creating very low water vapor concentrations and determining these with a reference instrument at exactly the same place and time. Given the limitations of this more established approach, using well-defined (higher) concentrations and a totally representative spectroscopic setup in combination with a very weak absorption line is arguably a highly valuable and reliable concept.

We are, however, well aware that this is a preliminary study we have performed using the tools available and it does not replace a more rigorous comparison in a more representative setting. This is again mentioned in the added section "Outlook".

Fig. 6 "A precision of 0.11 % at 1Hz is.." 0.11 % of what? Also Rˆ2 = 1.018 should either state the slope or the correlation coefficient I assume.

We have adapted the manuscript and now explicitly name these values "*relative Allan deviation*", i.e., relative to the measured mixing ratio; and "*coefficient of determination $R^2$*".

The calculation of the coefficient of determination is revised; the fact that the offset parameter is forced to remain zero is now taken into account in the calculation.

P13, L278 Figure 8(a) should be Figure 8

Done.

P16, L351 No, you were not successful to measure LS water vapor.

We have identified and explained a bias at high altitude, the system remained operational in the UTLS and there is no indication of inferior spectroscopic performance than in the troposphere, which we consider an outstanding achievement. However, we agree, that the term "successful" may be interpreted differently at this point. Therefore, we removed this terminology from the Conclusions section.